# Assessing Perceptual Load and Cognitive Load by Fixation-Related Information of Eye Movements

**DOI:** 10.3390/s22031187

**Published:** 2022-02-04

**Authors:** Jung-Chun Liu, Kuei-An Li, Su-Ling Yeh, Shao-Yi Chien

**Affiliations:** 1Department of Psychology, College of Science, National Taiwan University, Taipei 10617, Taiwan; b05901084@ntu.edu.tw; 2Department of Electrical Engineering, National Taiwan University, Taipei 10617, Taiwan; sychien@ganzin.com.tw; 3Ganzin Technology, Inc., Taipei 23141, Taiwan; kueian@ganzin.com.tw; 4Neurobiology and Cognitive Science Center, National Taiwan University, Taipei 10617, Taiwan; 5Center for Artificial Intelligence and Advanced Robotics, National Taiwan University, Taipei 10617, Taiwan; 6Graduate Institute of Brain and Mind Sciences, College of Medicine, National Taiwan University, Taipei 10617, Taiwan

**Keywords:** mental workload assessment, eye movement, perceptual load, cognitive load, load theory

## Abstract

Assessing mental workload is imperative for avoiding unintended negative consequences in critical situations such as driving and piloting. To evaluate mental workload, measures of eye movements have been adopted, but unequivocal results remain elusive, especially those related to fixation-related parameters. We aimed to resolve the discrepancy of previous results by differentiating two kinds of mental workload (perceptual load and cognitive load) and manipulated them independently using a modified video game. We found opposite effects of the two kinds of mental workload on fixation-related parameters: shorter fixation durations and more fixations when participants played an episode with high (vs. low) perceptual load, and longer fixation durations and fewer fixations when they played an episode with high (vs. low) cognitive load. Such opposite effects were in line with the load theory and demonstrated that fixation-related parameters can be used to index mental workload at different (perceptual and cognitive) stages of mental processing.

## 1. Introduction

Mental workload, the tuning between demands of the environment and the capacity of the individual [1], has a crucial influence on daily activities, especially when impaired performances can lead to negative consequences. For example, in a driving scenario where participants were asked to follow a vehicle with a concurrent secondary task (e.g., perform mathematics) [2], compared to when no such secondary task was required, the driver’s reaction time to step on the brake was delayed by about 0.5 s in response to the deceleration of the leading vehicle and the time-to-collision was shortened by about 1 s. The driver’s awareness of unexpected stimuli (e.g., a sudden appearing animal) or environmental changes (e.g., the change of the leading car’s position) was also reduced [3,4,5,6,7]. Such effects of mental workload on performances were also found in situations, such as aviation [8,9] and surgery [10], and in other sensory modalities, such as hearing [5,6,8] and touch [11]. Indeed, the mental workload has an unignorable impact on how we perceive the environment and others’ behaviors.

Assessing mental workload is therefore imperative because it provides us opportunities to avoid unwanted negative consequences. Take the above driving situation as an example, if the individual’s mental workload can be assessed, a system in the vehicle can be designed to receive such information and send a warning signal to the driver or even take over control if needed. Moreover, the vehicle or road designers can utilize such information to refine their products by, for example, removing any bad designs that may overload the user’s mental capacity.

Several types of signals were considered as valid indicators of an individual’s mental workload, among them cardiovascular signals, brain activities, and eye movements have received the most attention recently [12,13]. Cardiovascular signals are often measured in electrocardiography (ECG), which provides the cardiac cycles that are needed for assessing the mental workload. Electroencephalography (EEG) and event-related potential (ERP) are the most commonly measured brain activities for mental workload assessment. Unlike measures of ECG and EEG/ERP that rely on electrodes attached to the user’s body or brain, measures of eye movements are conducted by a remote or wearable device where images regarding the user’s eyes are captured by an infrared camera. Although each of these methods has had a sufficient evidence base for measuring mental workload, their validity depends on the scenario adopted [12,13]. We chose to use eye movements for assessing mental workload because it can effectively capture the variation of the mental workload regarding the visual demand of a task [13], which is one of the key demands in the scenarios mentioned above.

When assessing mental workload using eye-tracking devices, pupil size has been widely studied [12,13] since it is considered a useful index of mental effort or processing load that meets the three criteria proposed by [14]: sensitive to the within-task, between-task, and between-individual variations in processing load [15]. A positive correlation between the pupil size and the mental workload was demonstrated in several scenarios with different methods [15,16,17,18] (see also the section “Mental Effort and the Pupil” of [19]). Although pupil size is a valid indicator of mental workload, as mentioned by [15], it may reflect a basic physiological aspect of the mental workload, which is independent of the qualitative difference between the mental workload imposed by the task. Thus, with only the pupil size, practitioners can only be informed about the “intensity” of the mental workload without knowing where or at what stage the load is imposed on. For example, [16] manipulated the mental workload by changing the fallen speed of the blocks in Tetris, and in [17], the mental workload was manipulated by asking the participants to adopt an effortful strategy in a problem-solving task. Although the manipulations of the two studies imposed the workload at different processing stages (i.e., the perceptual and cognitive stage), both studies found pupil size increased when mental workload increased. Here, we propose that the fixation-related parameters of viewers’ eye movements can complement the pupil size measure and overcome such a deficiency. This is important for future mental workload assessment systems, since it can empower the system to deal with the mental workload directly on its source.

While fixation-related parameters of eye movements are also considered as useful indicators for mental workload assessment, no consensus has been reached yet [16,17,18,20,21,22]. Studies investigating the potential of using eye movements for assessing the mental workload often manipulated the mental workload either by adding a concurrent secondary task [18,20] or increasing the level of the task demand [16,17,21,22]. Although studies [16,17,18,20,21,22] suggested that fixation-related information can reflect the effect of mental workload, an opposite result pattern was found in different studies. For example, as the mental workload increased, the fixation duration was found to be either increased [17,18] or decreased [16,20,21,22].

Here, we hypothesize that the inconsistent results between previous studies using fixation-related parameters to assess mental workload are attributed to the fact that these studies may have assessed different types of mental workload. Based on the types of processing an individual relies upon, each task can impose a different level of load on our perceptual system (i.e., perceptual load) or higher-level cognitive mechanism (i.e., cognitive load), which determines what and how many items can be processed by the individual [23]. That is, if the perceptual requirement of the task exceeds the viewer’s perceptual capacity (i.e., high perceptual load), the perceptual load would serve as an early filter that only a few items the viewer sees can be taken for further processing. On the other hand, the cognitive load would mainly serve as a late filter that helps an individual to maintain his/her goal and response priority. More specifically, given that some cognitive resource is needed to maintain the goal of a task, if the task’s cognitive load is lower than the viewer’s cognitive capacity, the remaining cognitive resource would be further used to suppress the processing of task-irrelevant items. However, when the cognitive resource is exhausted by the task, both the relevant and irrelevant items would be considered and influence the following behavior/judgment. Therefore, different types of loads (perceptual vs. cognitive) have different effects on how the viewer processes and responds to the environment.

In line with this view, we assume that the studies where fixation durations were found to decrease as the mental workload increased may have involved the manipulation of perceptual load. For example, in [21], the mental workload was manipulated by adopting four types of roads, which were characterized by not only the complexity of the roads but also the number of items around or within the roads. Both characteristics imposed higher demand for perceptual processing than cognitive processing. Due to the high perceptual load, we hypothesize that only a few relevant items would be processed within a single fixation, which in turn may decrease the fixation duration and result in a higher fixation frequency to acquire the whole image of the task materials. In contrast, the studies where the fixation duration was found to increase with mental workload may have involved the manipulation of cognitive load. Take the study of [18] for example, a concurrent secondary task was adopted to manipulate mental workload. The secondary task asked the participants to imagine and manipulate an imaginary letter to answer a specific question, which mainly depended on cognitive processing. In such a case, due to the high cognitive load, we hypothesize that both items that are relevant or irrelevant to the main task (i.e., driving) are processed within a single fixation, which in turn may increase the fixation duration and decrease the fixation frequency because fewer fixations are enough to acquire the whole image.

The present study aimed to explore the potential of using fixation-related parameters for indexing the mental workload imposed on different processing stages. This not only helps resolve the discrepancy of results in previous studies but also expands the application of eye movement measures for mental workload assessment. As far as we know, this is the first study that addresses this issue. We believe that eye movements, especially fixation-related parameters, can reflect not only the general workload but also the effects of different types of mental loads (perceptual vs. cognitive). We modified a commercial game to manipulate perceptual load and cognitive load independently by changing the items within an episode or modulating the number of steps needed to pass the episode (and the difficulty in figuring out each step), respectively. We hypothesize that perceptual load and cognitive load have different effects on fixation-related parameters. More specifically, when perceptual load increases, fixation duration would decrease, whereas when cognitive load increases, fixation duration would increase. In addition, based on the assumption that the fixation duration indexing the number of items that can be processed during a given fixation, we further hypothesize that the fixation frequency (i.e., the number of fixations made during a single time interval) would show a reversed pattern against the fixation duration. That is, when perceptual load increases, fixation frequency would increase because participants would need to make more fixations to take all necessary items into the process. On the contrary, when cognitive load increases, fixation frequency would decrease since fewer fixations are enough to access all the items for further processing.

## 2. Materials and Method

### 2.1. Participants

The target number of participants was determined by an a priori power analysis with G*Power software 3.1.9.7 [24]. Effect sizes needed for this calculation were taken from three previous relevant studies, as well as our pilot study that had the same experimental procedure as in the current study except for the presentation order of the conditions and the way to familiarize participants with the rules of the game. Effect sizes for fixation duration in the three previous studies used for estimating the sample size here were taken from significant main effects of mental workload (η_p_^2^ = 0.87 for [16], 0.52 or 0.38 for [18] where participants drove on the road or highway, respectively, and 0.37 for [21]). The effect size values were either reported in these studies or estimated by the F value and degree of freedom [25]. For our pilot study, we calculated the effect sizes for main effects of perceptual load on fixation frequency (η_p_^2^ = 0.86) and fixation duration (η_p_^2^ = 0.93), and those for main effects of cognitive load on fixation frequency (η_p_^2^ = 0.71) and fixation duration (η_p_^2^ = 0.76). According to the power analysis, 3–11 participants were required to reach the statistical power of 0.8.

We recruited nine undergraduate students (6 males and 3 females; mean age = 23.4 ± 4.36, ranged from 20 to 34 years) from National Taiwan University. All participants reported corrected-to-normal vision and were native speakers of Mandarin Chinese. Each participant gave informed consent to participate in the experiment and was paid $20 after s/he completed this 3-h long experiment. This study was approved by the Research Ethics Committee at National Taiwan University (NTU ERC: 202007EM029) and was implemented accordingly.

### 2.2. Equipment

The experiment was run on a desktop computer with a screen in front of the participant. The screen was set approximately 57 cm from the eyes of participants, which made up a 40° × 50° field of view. An EyeLink 1000 eye tracker sampled at 500 Hz was placed below the screen to collect eye movement data from participants’ dominant eye. The standard nine points calibration and validation procedure were provided at the onset of each block including both the blocks of practice and experiment phase, which repeated until the average accuracies of the gaze samples were lower than 1°.

### 2.3. Task

*Baba Is You* [26] is a puzzle-solving video game similar to the classical game Sokoban [27], which is considered one of the best puzzle games in recent years [28] and is becoming popular worldwide. We chose this game based on its popularity, which indicates a high ecological validity, and its nature that can be used to modify the two types of mental workload independently (see Section 2.4 for more details). Unlike traditional puzzle games where the rules of the game are fixed, in *Baba Is You* the rules are also a part of the game that can be changed by the players. More specifically, in *Baba Is You*, each puzzle contains several objects and corresponding words that describe the rules. To solve the puzzle, the player should change and create new rules by rearranging the words. The affordance of the object can be dissociated from its actual function. For example, the affordance of an object “wall” indicates that it would stop anything trying to pass through. However, if this function is not described by the words, the player can control his/her character to pass through the object “wall”. Moreover, other functions irrelevant to the wall may also be assigned to the object “wall” if it is described by the word.

In the current study, a modified version of *Baba Is You* was adopted and programmed by Python3 to create customized episodes. Figure 1 depicts one of the sample episodes, which was adopted to introduce participants to the basic rules of the game. As shown in this example, several word tiles and objects are presented on the game field. The word tiles with a Chinese word in each of them can be further categorized into three types based on their functions: The colorful square word tiles are nouns, each of which corresponds to a particular type of object in the game field. The circle word tiles are descriptors, each of which indicates a specific property in the game that needs to be linked to the object to exert its effect. Finally, the dark gray square word tiles with a white character on them are connectors that link the nouns and the descriptors together. The three types of word tiles would arrange in a form similar to an equation that spells out the rules in the current episode. For example, as the equation shown in the top-right corner of Figure 1, the light gray square word tile 牆 (meaning “wall”) is a noun, which corresponds to the column of walls in the middle of the game field; the circle word tile 止 (meaning “stop”) is a descriptor, which indicates that the objects assigned cannot be passed through by any other objects or word tiles, and the square word tiles “=” is a connector, which links the light gray square word tile 牆 with the circle word tile 止. The alignment of these word tiles would create the rule that all objects and word tiles cannot pass through the column of walls in the middle of the game field. The objects (i.e., the flag, rock, wall, and the blank pink square) are the background of the episode that the player cannot interact with if no descriptor is linked with the nouns corresponding to the object. In each episode, the player aims to find a solution that makes the objects controlled by the player to touch the object whose corresponding noun linked with the descriptor 勝 (meaning “win”).

Figure 2 illustrates a solution regarding the episode depicted in Figure 1. A player can first move the blank pink square along the route indicated by the dotted arrow to push the word tile 旗 (meaning “flag”) and change the equation made by the word tiles (the top panel in Figure 2). The new equation indicates that the player can now control the flag on the right side of the game field, and the player can further move the flag to touch the stone to win the game (the middle and bottom panels in Figure 2). All the operations are executed by using the arrow keys in the keyboard, and in some cases where the episode may reach a dead-end (e.g., no objects are connected with the descriptor 你), an additional z key that can restore the game to the status of previous steps is provided to the participants.

### 2.4. Workload Manipulation

The modified game was adopted mainly because it allowed us to manipulate perceptual load and cognitive load independently. As shown in Figure 3 (compare the episodes in the top panels to those in the bottom panels), the perceptual load was manipulated by changing the number of objects or word tiles presented in the game field without modifying the position and identity of the key elements, and thus, the rules were maintained. Figure 3 also depicts an example of the manipulation of cognitive load where only one noun differed between the episode in the left panel and that in the right panel (i.e., the noun “旗” in the low cognitive load condition was changed into “它” in the high cognitive load condition). In this case, due to the change of the key elements, the number of necessary steps was increased in the high cognitive load condition, which made the solution of this episode not as intuitive as that in the low cognitive load condition. That is, to figure out these steps, participants had to gain insight into the layout of the episode in the high cognitive load condition.

Figure 4 depicts the average number of items of the four mental workload conditions adopted in this study, which indicates a significantly higher number of items in the high perceptual load condition (F(1,7) = 52.06, *p* < 0.001). On the other hand, the cognitive load was manipulated by changing the position or the identity of the key elements without changing the number of items presented in the game field; thus, there was no statistical difference in the number of items between the high and low cognitive load conditions (F(1,7) = 3.54, *p* = 0.1).

To ensure the validity of the manipulation, especially the manipulation of cognitive load, the NASA Task Load Index (NASA-TLX) was adopted to evaluate the subjective workload under each condition. NASA-TLX is a tool widely used in studies that investigate the effect of workload [29,30]. It contains six rating questions (0–100) in six dimensions to assess participants’ subjective workload, including mental demand, physical demand, temporal demand, performance, effort, and frustration [29,30]. Because the contribution of each dimension to the subjective workload differed, different weights were added to each dimension when calculating the total score of NASA-TLX. The following formula depicts the way that the total score of the NASA-TLX was computed:(1)$${Score}_{total}=\overset{6}{\underset{i=1}{\sum }}{Weight}_{i}\times {Score}_{i}$$
where *Score_i_* indicates the raw score of the specified dimension, and *Weight_i_* indicates the weight given to the specified dimension, which is calculated by a comparison task conducted by each participant. In the comparison task, two words indicating two of the six dimensions were presented in each trial, and participants were asked to judge which of the two dimensions contributed more to the subjective workload. Fifteen pairwise comparisons were made, and the weight of each dimension for each participant was calculated by the following formula:(2)$${Weight}_{i}=\frac{number\;of\;wins}{Total\;number\;of\;Comparisons}$$
where the *number of wins* indicates the number of times the specified dimension won the comparison, and the *Total number of Comparisons* indicates the total number of comparisons in the comparison task (i.e., 15).

### 2.5. Design

A within-subject 2 × 2 factorial design with two levels of cognitive load (high, low) and two levels of perceptual load (high, low) was adopted, resulting in four mental workload conditions. In each condition, eight episodes were created in which three episodes were used in the practice phase and five episodes in the experiment phase. The experiment consisted of two 1.5–2 h sessions with an interval longer than 6 h between the two sessions. All the factors and episodes were counterbalanced to prevent or minimize possible carry-over effects. The two perceptual load conditions were conducted in the two sessions respectively. Half of the participants who conducted the episodes with high perceptual load in the first session would conduct a low perceptual load version in the second session, while the other half of participants would conduct the episodes in the reverse order. The two cognitive load conditions were interwoven in each session, and the order of these two conditions for each episode was counterbalanced across the two sessions.

### 2.6. Procedure

Figure 5 depicts the procedure of the sessions. Each session was divided into three parts. In the first part, participants were asked to complete 15 simple episodes, which introduced participants to the basic rules of the task. If participants failed to pass all the episodes in this part, the experimenter would pick up the episode in which the participants failed and re-introduce the rules indicated by the episode and the way to pass the episode. Then, participants were asked to conduct again the episodes that they had failed. The above procedure would repeat until participants could pass all 15 episodes ensuring that participants were familiar with the basic rules regarding the task. After the introduction phase, to be familiar with the task, participants were given four blocks of three episodes where each block corresponded to one of the mental workload conditions. After each block, participants were asked to complete the NASA-TLX. Finally, in the experiment phase, participants were asked to complete a block of 10 episodes after which the NASA-TLX was conducted again, followed by the comparison task of the NASA-TLX mentioned in Section 2.4. In the practice phase and the experiment phase, each episode had a five-minute time limit, and participants were asked to pass the episode using as few steps as possible. After the participants passed the episode or reached the time limit, the episode would be terminated and the participate moved forward to the next episode.

### 2.7. Statistical Analysis

Data were analyzed using JASP version 0.15 [31] as well as the ARTool [32], if necessary. First, to evaluate the validity of the load manipulation, a two-way repeated measure ANOVA was conducted on the NASA-TLX scores to verify the manipulation of the two types of loads on the subjective workload assessment. As described in the previous section, a total of five NASA-TLXs were conducted. The four NASA-TLXs in the practice phase, which corresponded to the four mental workload conditions respectively were subject to this ANOVA. The NASA-TLX conducted at the end of the experiment phase was used to evaluate the general subjective workload, and no participant reported the subjective workload higher or lower than the three standard deviations from the mean. Second, three types of data were used to index participants’ performance on the game, including the total number of steps used to pass the episodes, the number of times that participants encountered a dead-end, and the ratio of passing the episodes (accuracy). A two-way repeated measure ANOVA was conducted on the manual response data to examine the effect of the two types of loads. Finally, for the eye movement data, two types of fixation-related parameters were analyzed. Fixation duration refers to the average duration of fixations that participants made during the episode, which can be calculated with the following equation:(3)$$\mathrm{Fixation}\;\mathrm{duration}=\frac{\sum_{i=1}^{n}{dur}_{i}}{n}$$
where *n* indicates the number of fixations made during the episode, and *dur_i_* indicates the duration of the ith fixation. Fixation frequency indicates the number of fixations made per second during the episode, which can be computed with the equation:(4)$$\mathrm{Fixation}\;\mathrm{frequency}=\frac{n}{{dur}_{task}}$$
where *n* indicates the number of fixations made during the episode, and *dur_task_* indicates the duration of the episode in seconds. All fixations were detected by the algorithm in EyeLink 1000. To avoid possible artifacts, an additional filter was set to filter out the fixations detected with durations shorter than 80 ms. A two-way repeated-measures ANOVA was conducted on the fixation-related parameters to examine the effects of the two types of loads.

## 3. Results

### 3.1. NASA-TLX Score during the Practice Phase

Figure 6 depicts the NASA-TLX score in different mental workload conditions. Participants reported higher subjective workload in the high cognitive load condition compared to the low cognitive load condition (M = 54.95 ± 4.2 and 34.84 ± 3.7 for high and low cognitive load conditions, respectively), F(1,8) = 21.3, *p* < 0.01, η_p_^2^ = 0.73, which provides evidence validating the effect of load manipulations, especially the manipulation of cognitive load, in this study.

### 3.2. Performance of the Episode

All the behavior indexes (i.e., number of steps needed to pass, number of dead-ends encountered, and accuracy) were impacted by cognitive load (Figure 7). Compared to the low cognitive load condition, when encountered an episode with a higher cognitive load, participants executed more steps (M = 359.5 ± 38.1 and 163.1 ± 22.7 for high and low cognitive load conditions, respectively, F(1,8) = 55.17, *p* < 0.001, η_p_^2^ = 0.87), encountered more dead-ends (M = 4.1 ± 0.6 and 1.4 ± 0.4 for high and low cognitive load conditions, respectively, F(1,8) = 41.5, *p* < 0.001, η_p_^2^ = 0.84) and passed the episode with a lower accuracy (M = 0.38 ± 0.08 and 0.91 ± 0.05 for high and low cognitive load conditions, respectively, F(1,8) = 55.4, *p* < 0.001, η_p_^2^ = 0.87). However, no effect of perceptual load or interaction on behavior indexes was found, ps > 0.05. Combined with the result of NASA-TLX, these results imply that behavior data can only reflect the type of load that had the highest contribution to the perceived mental workload.

### 3.3. Eye Movement Data

Both perceptual load and cognitive load impacted the fixation-related parameters (Figure 8). Higher perceptual load led to decreased fixation durations (M = 312.3 ± 16.5 ms and 330.3 ± 13.9 for high and low perceptual load conditions, respectively, F(1,8) = 5.9, *p* < 0.05, η_p_^2^ = 0.42), but higher cognitive load led to increased fixation durations (M = 329 ± 15.2 ms and 313.6 ± 14.5 for high and low cognitive load conditions, respectively, F(1,8) = 18.6, *p* < 0.01, η_p_^2^ = 0.7). In contrast, higher perceptual load led to increased fixation frequencies (M = 2.61 ± 0.08 times/second and 2.48 ± 0.08 times/second for high and low perceptual load conditions, respectively, F(1,8) = 7, *p* < 0.05, η_p_^2^ = 0.47), but higher cognitive load led to decreased fixation frequencies (M = 2.47 ± 0.07 times/second and 2.62 ± 0.09 times/second for high and low cognitive load conditions, respectively, F(1,8) = 20.9, *p* < 0.01, η_p_^2^ = 0.72).

## 4. Discussion

We have shown here as proof of concept that eye movement signals can reflect the effects of different types of mental loads (i.e., perceptual load and cognitive load) imposed by the current task. By modifying both the perceptual and cognitive processing requirements of a commercial game *Baba Is You* and manipulating perceptual load and cognitive load independently, our results revealed that perceptual load and cognitive load had an opposite effect on fixation-related parameters. These results support our hypotheses: Fixation duration decreased as the perceptual load increased and increased when the cognitive load increased while fixation frequency increased as the perceptual load increased and decreased as the cognitive load increased. The differentiation of the two kinds of mental workload can only be seen with eye movement signals, but not behavioral results (accuracy, number of steps, and number of dead-ends) and questionnaires such as NASA-TLX. As far as we know, this is the first study demonstrating the possibility of using fixation-related parameters (i.e., fixation duration and fixation frequency) for differentiating and assessing the two kinds of mental workload (perceptual and cognitive load) separately.

Such opposite effects on fixation-related parameters are consistent with the load theory [23] wherein perceptual load and cognitive load had opposite effects on determining whether an item could be taken into further processing. As perceptual load and cognitive load serve as early and late filters respectively, only a portion of items can be taken into further processing with high perceptual load, whereas both relevant and irrelevant items would be further processed with higher cognitive load. Therefore, when conducting a task, both perceptual load and cognitive load work together to determine how many items could be taken into further processing. Moreover, with a higher perceptual load, only a few items can be taken for further processing at each fixation. Thus, the fixation duration can be shortened, and participants need more fixations to receive a whole representation of the game materials. In contrast, when cognitive load is high, because of the failure to suppress irrelevant items for further processing, the number of items needed to be processed within a fixation is larger than what would be expected when cognitive load is low. Therefore, fixation duration would be lengthened, and participants can obtain the whole image with fewer fixations.

The opposite effects of different types of loads on fixation-related parameters we found here suggest that eye movement parameters, especially the fixation-related parameters, can reveal not only more than the general mental workload perceived as assessed by other measures (e.g., behavior performances such as the total steps used and questionnaires such as NASA-TLX) but also the level of different types of loads perceived by the individual. Such information may be useful in several application domains. For example, in the applications regarding driving safety, several studies investigated the effect of mental workload on driving behavior, especially the awareness of and the response to unexpected objects (e.g., pedestrian) or events (e.g., sudden braking of a leading car), which are critical for driving safety and suggest that high mental workload has negative effects on driving [3,4,5,6,7,33]. Although an index of general workload can help us avoid immediate hazards, it may not be helpful to reduce the mental workload perceived because the system that receives the information to assist the user (e.g., the driver monitor system in the vehicle) does not know the sources of the load. If the system can be informed of the sources of load by utilizing eye movement information, it can further help the driver to reduce the mental workload and diminish the risk in the remaining journey. For example, the system can reduce the demand for perceptual processing by reducing the information provided on the vehicle as much as possible. In addition, the system can also provide some advice regarding the selection of the route or assist with controlling the vehicle to reduce the demand for cognitive processing. In addition to the utilization in a real-time manner, such information is also worthy in designing the vehicles or systems because it provides us a clear index about whether the demands for either perceptual processing or cognitive processing of the product exceed the capacity of the users.

In addition to the usage of the critical situation as mentioned above (e.g., driving), such eye movement information is also valuable to education or training. According to the cognitive load theory in education [34], three different loads can be perceived in learning: (1) the intrinsic cognitive load, which is imposed by what a student is learning; (2) the extraneous cognitive load, which is imposed by the teaching materials and the activities that the student needs to interact with; (3) the germane cognitive load, which is imposed by the amount of effort that the student has to put into to process the information relevant to the learned materials. Thus, good learning or training needs to maximize the germane cognitive load and minimize the extraneous cognitive load.

According to the load theory [23], the combined effect of perceptual load and cognitive load on fixation-related parameters can be considered as a signal that tells us about how much relevant and irrelevant information would be taken into further processing while doing a task. In line with this view, when the student suffers from the high perceptual load as indicated by a shorter fixation duration and/or higher fixation frequency, it implies that only a small portion of relevant information can be processed at each fixation. The student thus needs to make more fixations to obtain a whole image of the teaching materials, which increases the activities that the student needs to interact with the materials, namely a higher extraneous cognitive load. Similarly, when the student undergoes high cognitive load as expressed by a longer fixation duration and/or lower fixation frequency, it is inferred that the cognitive resource of the student may be exhausted, and no resource can be used to suppress the processing of the irrelevant information. Thus, more effort needs to be put into the irrelevant information, also indicating a higher extraneous cognitive load.

Therefore, eye movement information may indicate how much effort students put into irrelevant information or redundant behaviors (i.e., extraneous cognitive load). More importantly, it indicates where this unwanted load comes from, which in turn provides a guide for the teachers to improve their instructional effectiveness. For example, if the teaching material is hard to understand and students show shorter fixation durations or higher fixation frequencies when interacting with this material, it may hint to the teacher that there is too much information presented (i.e., higher perceptual load). However, if students show longer fixation durations or lower fixation frequencies when interacting with the material, it may indicate that the students do not realize the key information of the material, and thus, they spend more time organizing the information received and/or deciding where to explore next. In both cases, to improve instructional effectiveness, the teacher can redesign the teaching material and make it more concise and organized. For the latter case, in addition to refining the material, the teacher may need to put more efforts into clarifying the information presented and teaching the students strategies to interact with the material in an efficient way, such as an efficient search pattern, that helps the student attain the key information with ease.

Finally, in this study, the eye movement information was measured by a remote eye tracking device; however, in the application situations as mentioned above, such a device may not be the best choice. Although the remote eye-tracking device has a higher temporal resolution than other types of eye-tracking devices (i.e., wearable devices), it also sets a constraint on the movement of the user; for example, the device used in the current study only allows the participants to move their head within a 22 cm × 18 cm × 20 cm square box. Moreover, a space is needed in the environment to set up the device. Thus, it is not suitable for applications where an individual needs to walk around and where no extra space can be used to set up the remote device (e.g., riding a motorcycle). In addition, high temporal resolution is unnecessary for the abovementioned applications. That is, an eye-tracking device with a temporal resolution higher than 60 Hz is enough to measure the eye movement parameters adopted in this study (i.e., fixation duration and fixation frequency). Therefore, to maximize the value of eye movement information in assessing the mental workload, future studies are recommended to adopt wearable eye-tracking devices in real application situations to further verify the results we found here.

## 5. Conclusions

Assessing mental workload is imperative for avoiding unwanted negative consequences (e.g., car crashes and poor instructional effectiveness) in several daily activities (e.g., driving and learning). As far as we know, this is the first study that demonstrated the potential of using fixation-related parameters for differentiating and assessing the two kinds of mental workload (perceptual and cognitive load). That is, perceptual load and cognitive load had opposite effects on both the fixation duration and fixation frequency. Future studies should consider these results when developing an algorithm or system for assessing the mental workload based on eye movements. This information may empower the algorithm or system to better deal with the source of the mental workload that not only reduces the mental workload perceived by the user but also reduces the possibility that the user may be overloaded again. Educators can also use this information as guidance to modify the materials in teaching and learning scenarios.

## Figures and Tables

**Figure 1 sensors-22-01187-f001:**
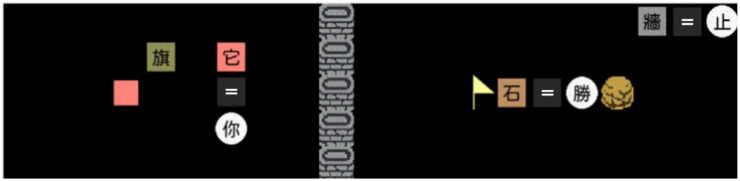
An example of the game field where several objects and word tiles are presented, which determine the layout and the rules of the episode.

**Figure 2 sensors-22-01187-f002:**
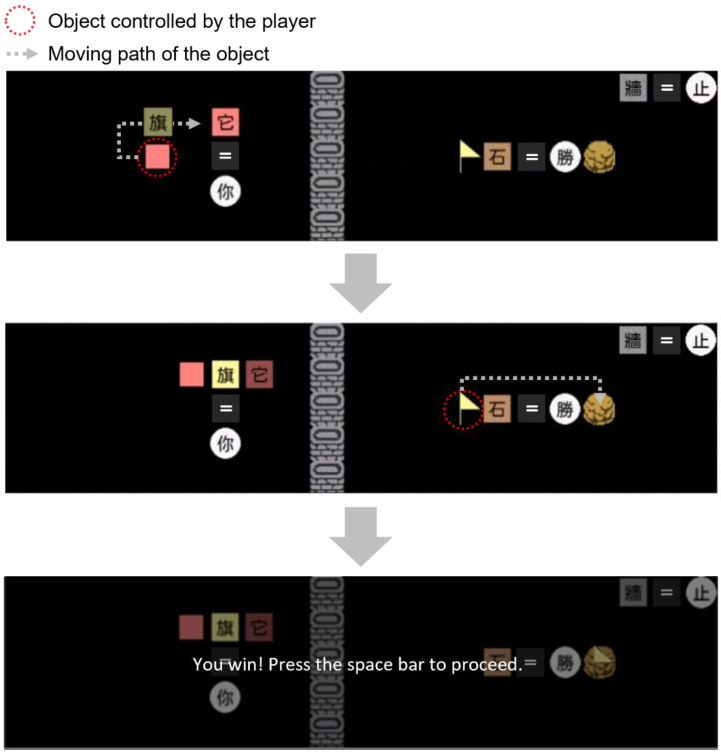
An example of a solution for the episode depicted in Figure 1. At first ((**top**) panel), a player moves the blank pink square to modify the equation made by the word tiles on the left side of the game field, resulting in changing the object controlled by the player (as depicted in the (**middle**) panel). Later, the player moves the flag to touch the stone to win the game (as depicted in the (**middle**) and the (**bottom**) panels). The red dotted circle indicates the object controlled by the player, and the white dotted arrow indicates the path along which the player moves the object; both are not shown in the game episode (see video: http://epa.psy.ntu.edu.tw/Liu_etal_2022_sensors_demo_video.mov accessed: 22 January 2022).

**Figure 3 sensors-22-01187-f003:**
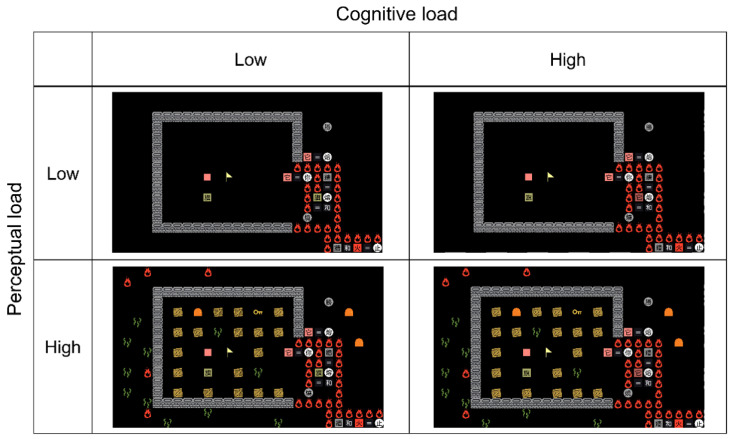
An example of the four mental workload conditions adopted in the experiment.

**Figure 4 sensors-22-01187-f004:**
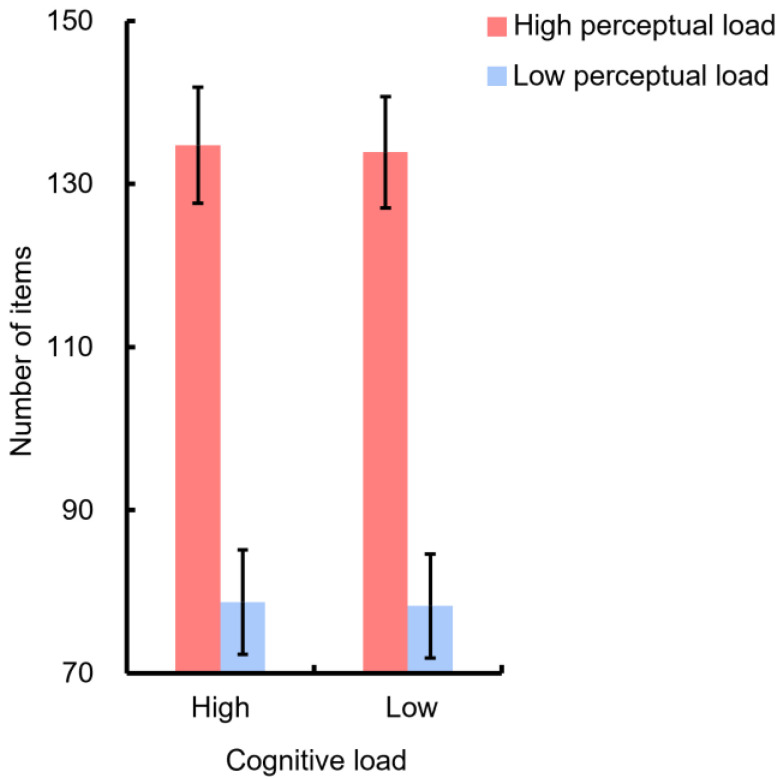
The average number of items (including the objects and word tiles) presented in the four mental workload conditions. Error bars indicate one standard error from the mean.

**Figure 5 sensors-22-01187-f005:**
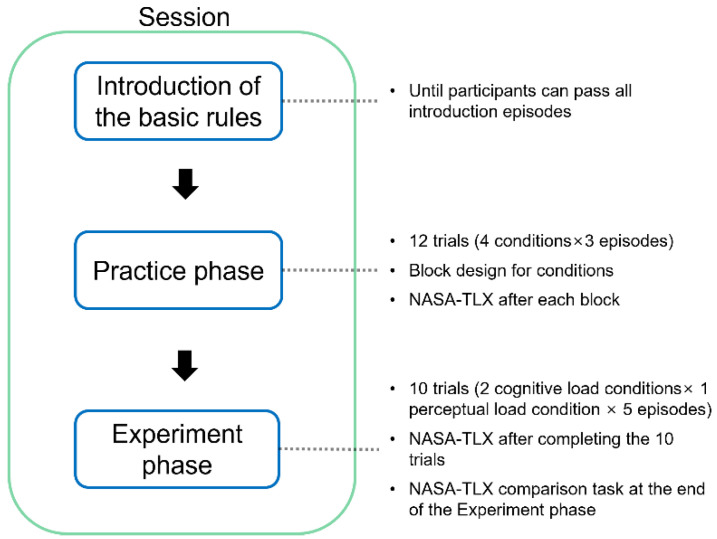
The procedure of one session in the experiment.

**Figure 6 sensors-22-01187-f006:**
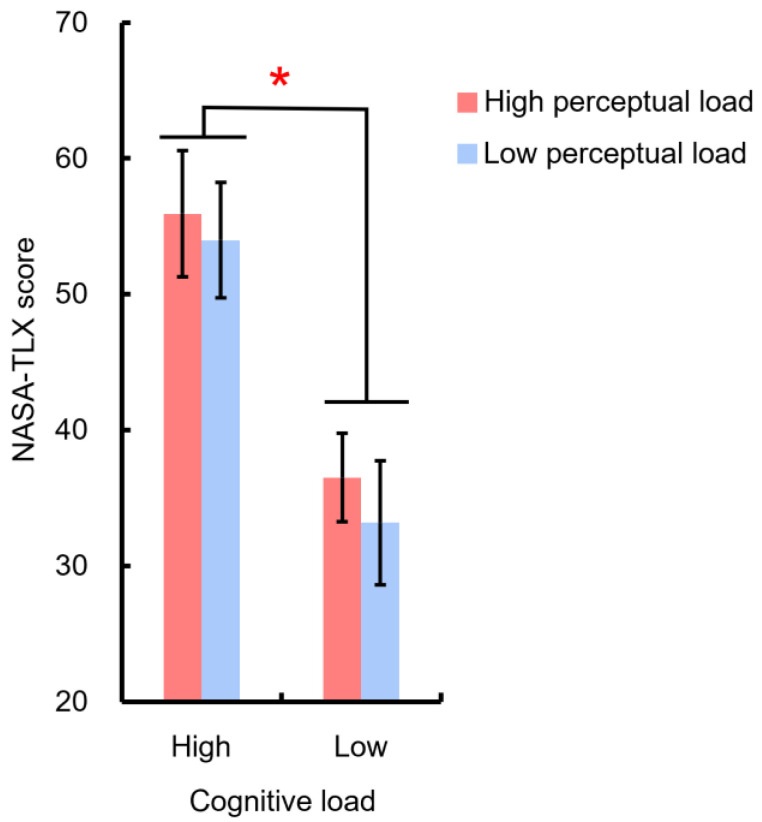
NASA-TLX scores differed in different mental workload conditions. Error bars indicate one standard error from the mean. * *p* < 0.05.

**Figure 7 sensors-22-01187-f007:**
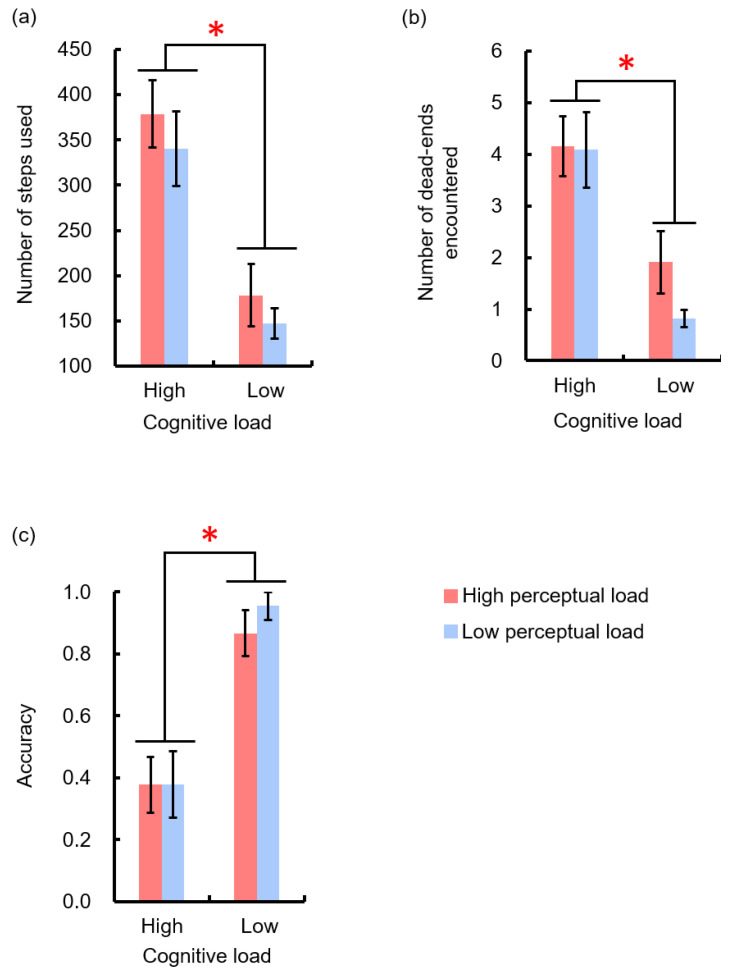
Behavior performance in episodes with the different mental workloads. When encountering episodes with a higher cognitive load, participants (**a**) needed more steps to pass the episode, (**b**) encountered a higher number of dead-ends, and (**c**) had a lower accuracy. Error bars refer to one standard error from the mean. * *p* < 0.05.

**Figure 8 sensors-22-01187-f008:**
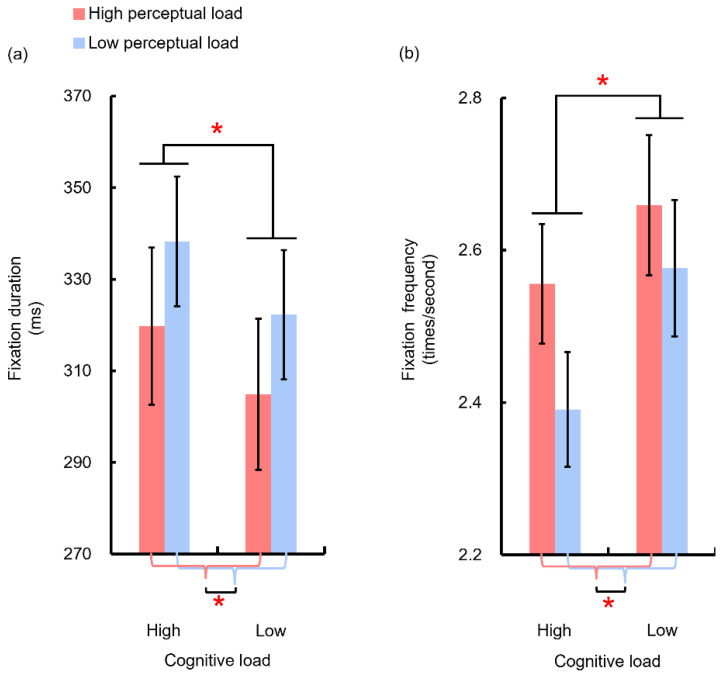
Perceptual load and cognitive load had an opposite effect on fixation duration and fixation frequency. (**a**) Higher perceptual load led to decreased fixation durations but higher cognitive load led to increased fixation durations. (**b**) Higher perceptual load led to increased fixation frequencies but higher cognitive load led to decreased fixation frequencies. The curly brackets and the asterisk at the bottom of each graph indicate the main effect of the perceptual load where the red and blue curly brackets indicate the mean value of the two red lines and the two blue lines, respectively, and the asterisk at the bottom refers to a significant difference between these two mean values. Error bars refer to one standard error from the mean. * *p* < 0.05.

## Data Availability

The data supporting this study’s findings are available from the following link: https://osf.io/ukt46/?view_only=f1c95e22cabc4d9bbef7f9268909960c (accessed: 23 December 2021).

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
