# Peer review of "Assessing Perceptual Load and Cognitive Load by Fixation-Related Information of Eye Movements"

_sensors, 2022, doi:10.3390/s22031187_

Round 1

Reviewer 1 Report

The usage of video games to extract information related to  mental workload (perceptual load and cognitive load) represent an important subject and the paper focus and presents interesting methods in the field.

Considering the sensor topics the paper might be improved and related to sensing of eye tracking underlining the capability of the used system and a comparison with other systems highlighting the selection reason. 

How the game was chosen and why? How the data was extracted and what bout data storage and processing. Exist any contribution on software development or the contribution is on data analysis. 

The number of participants is quite reduced and this number might be justified. It is recomended an extended number of participants (e.g. 20 participants). A comparison with exiting reported technics might be included in the paper.

The conclusion might be improved highlighting the originality and the novelty of the proposed method based on eye tracking. What about the robustness of the method.

The conclusion might be revised highlighting the originality of the approach.

Reviewer 2 Report

Some issues should be clarified before this paper can be considered accepted for publication.

1) Strong descriptions to address the importance of the proposed topic should be given. Authors are also expected to highlight the significance of this work.

2) Then, a comparison among similar works should be given. It is especially important to make a summary of those existing research articles and point out what and how this newly proposed work sounds.

3) Details in regard to the implementation of this work should be given instead of the architecture.

4) Statistics results could be improved by adding more details in regard to the experiment design and the performance with other similar works.

5) The presentation of the work, especially the English part, could be checked and improved.

Reviewer 3 Report

The paper “Assessing perceptual load and cognitive load by fixation-related information of eye movements“ describes the eye-tracking study performed on eight participants playing the game “Baba is you”.

Although the issue of cognitive/perceptual load measurement is definitely interesting, I am not satisfied with the paper. From my point of view, the study might serve as a pilot or proof of concept. The results are based on the statistic evaluation of eye-movement metrics and NASA TLX and I think that the number of nine participants is simply not enough for such type of analysis. Could authors provide test power analysis? I can understand that for eye-tracking research, it might be problematic to run experiments on very high numbers of participants, but nine is a really small number.

My advice is to use this experiment as a pilot for a larger study, which results might be then published in a high-impact paper.

I tried to understand the principle of the game from the description in section 2.3, I have read that many times and I had no idea. It might be beneficial to add a link to some tutorial videos.

I am also missing deeper insight into the issue of cognitive load measurement. For example, on the basis of pupil size…

Generally, I like to asterisks marking significant values in the boxplots, however, the system used by the authors is not clear to me. For example, Figure 8 – what does the asterisk at the bottom represents? Difference between the blue ones? Red ones? Red vs. Blue? Red and Blue?

Moreover, I think that the differences in fixation frequency/fixation duration between high and low perceptual load are caused by the different number of objects in the stimuli. With a higher number of objects, there will be higher frequency and lower fixation duration.

Line 376 – mismatched reference

Round 2

Reviewer 3 Report

ok